# Psychometric properties of the Generalised Anxiety Disorder Dimensional Scale in an Australian sample

David Groves[1], Theodora Binasis[1], Bethany Wootton[2], Karen Moses[1]*

**1** School of Psychology, Western Sydney University, Penrith, New South Wales, Australia, **2** Discipline of Psychology, Graduate School of Health, University of Technology Sydney, Ultimo, New South Wales, Australia

* K.Moses@westernsydney.edu.au

**Data Availability Statement:** All relevant data are within the paper and its Supporting Information files.

**Funding:** The authors received no specific funding for this work.

## Abstract

The Generalised Anxiety Disorder Dimensional Scale is a new measure of generalised anxiety disorder developed to assist clinicians in the dimensional assessment of generalised anxiety disorder by the Diagnostic and Statistical Manual (Fifth Edition) Anxiety, Obsessive-Compulsive Spectrum, Posttraumatic, and Dissociative Disorder Work Group. This study aims to evaluate the psychometric properties of the scale in an Australian community sample. A sample of 293 Australians (72.7% female) aged between 18 and 73 ($M = 28.31$ years; $SD = 12.11$ years) was recruited. Participants completed the Generalised Anxiety Disorder Dimensional Scale, as well as related measures used to assess convergent and discriminant validity. A small proportion of the sample ($n = 21$) completed the scale a second time to assess test-retest reliability. The scale demonstrated a unidimensional factor structure, good internal consistency (Cronbach's $\alpha = .94$), good test-retest reliability (ICC = .85), good convergent validity with the Generalised Anxiety Disorder– 7 item ($r_s = .77$), and discriminant validity with the Panic Disorder Severity Scale–Self Report ($r_s = .63$). The scale appears to be a reliable and valid measure of generalised anxiety disorder symptomology for use in the Australian population.

## 1. Introduction

Generalised anxiety disorder (GAD) is characterised by worry and/or anxiety about multiple activities or events that is excessive and difficult to control [1]. International epidemiological research estimated that GAD has lifetime prevalence of 3.7% and 12-month prevalence of 1.8% [2]. In comparison, the Australian population has been found to have a higher than average rate of GAD, both in one-year prevalence (3.6%) and lifetime prevalence (8%) [2]. The disorder is associated with reduced quality of life [3] and imposes a significant burden [4].

Fortunately, effective treatments for GAD exist, which are best utilized after an evidence-based clinical assessment [5]. Self-report measures are considered to be a key component of this evidence-based assessment and provide a number of benefits, including that they can be used to inform diagnosis [6] and monitor symptom change over time [7]. Whilst the prevailing

**Competing interests:** The authors have declared that no competing interests exist.

diagnostic approach has been the categorical classification of mental disorder, the DSM-5 introduced a supplemental dimensional approach. A dimensional approach to disorder classification allows for a better assessment of patterns, severity and number of symptoms experienced [8].

The DSM-5 Anxiety, Obsessive-Compulsive Spectrum, Posttraumatic, and Dissociative Disorder Work Group developed dimensional scales for the DSM-5 anxiety disorders [9]. Using a common, 10-item template, these scales assess the constructs of fear and anxiety that are shared–but manifest differently–across each of the disorders [10]. To date, the psychometric properties of the Dimensional Anxiety Scales have been examined in clinical and community samples in the USA [9], Germany [10, 11], the Netherlands [12, 13], and Brazil [14]. However, no studies have examined the properties of the dimensional anxiety scales in an Australian sample. Furthermore, only one of the studies mentioned above was conducted using the English version of the scales [9]; in other studies, the scales were translated into German, Dutch, and Portuguese-Brazilian.

Across studies, the Generalised Anxiety Disorder Dimensional Scale (GAD-D) has been found to have a unidimensional structure [10, 11, 14], good-to-excellent internal consistency [9–12, 14] and adequate test-retest reliability (intraclass correlation coefficients between 0.54 and 0.70; [18, 23]). Correlations of the GAD-D with measures of similar and different constructs have provided support for both convergent [15] and divergent validity [9–11]. Finally, DeSousa et al. [14] separated their sample into male and female populations and found good internal consistency for each, indicating that reliability was not affected by participant gender.

Taken together, research suggests that the GAD-D is a valid and reliable measure of GAD severity, that it is a freely available, brief and easy to administer. Despite research undertaken to date, there a number of gaps within the existing literature which are important to address. Firstly, an evaluation within an Australian sample is currently lacking. Secondly, only one study has examined the psychometric properties of the English version [9]. Thirdly, studies conducted to date have largely been homogenous in sample characteristics [9, 11]. Fourthly, studies have not uniformly tested each psychometric properties of the GAD-D. For example, only Beesdo-Baum et al. [10] and DeSousa et al. [14] established unidimensionality for the scale. Consequently, the factor structure of the English version of the GAD-D has not yet been examined. Finally, test-retest reliability was only assessed in American [9], Brazilian [14], and German [11] samples, and as mentioned above, these studies yielded varying results. Given the above, the generalizability of the GAD-D, particularly within the general population and the Australian context more broadly, remains unknown. Given that the prevalence [2] and expression [16] of GAD-D varies across countries, an evaluation within other cultures, including the Australian context is warranted.

The present study aims to measure the psychometric properties of the GAD-D in a community sample of Australian adults. Establishing these psychometric properties using an Australian sample would give clinicians working in Australia an empirical basis for using this scale to screen for, measure, and monitor GAD symptomology and severity in the populations they work with and will also add to the growing literature demonstrating the psychometric properties of the DSM-5 dimensional scales in the Australian population [17–20]. The following were investigated: 1) factor structure; 2) reliability; and 3) convergent and discriminant validity. It was hypothesised that the psychometric properties of this scale will be consistent with previous evaluations [9, 10, 12, 14]. Specifically, the GAD-D is expected to demonstrate a unidimensional factor structure, excellent internal consistency, good test-retest reliability, convergent and discriminant validity.

## 2. Methods

### 2.1. Participants

A total of 355 participants commenced this study. Participants who met exclusion criteria or provided incomplete data were removed from all analyses. Two hundred and ninety-three Australian adults ($M$ = 28.31 years; $SD$ = 12.11 years) were included in the final sample. The sample was predominantly female (72.7%). To be included in this study, participants were required to 1) be aged 18 years or over, 2) live in Australia, and 3) be fluent in English. Further sample characteristics are presented in Table 1. As this study utilized a non-clinical community sample, psychiatric history was not sought from participants. Ethical approval was granted by the Western Sydney University Human Research Ethics Committee.

**Table 1. Participant characteristics (N = 293).**

| Variable | *n* | % |
|---|---|---|
| **Gender (% female)** | **213** | **72.7** |
| Marital Status | | |
| Single | 186 | 63.5 |
| Married | 53 | 18.1 |
| De Facto | 38 | 13.0 |
| Divorced | 11 | 3.8 |
| Widowed | 1 | 0.3 |
| Separated | 4 | 1.4 |
| Employment Status | | |
| Working part time | 83 | 28.3 |
| Working full time | 80 | 27.3 |
| Unemployed | 13 | 4.4 |
| Studying | 101 | 34.5 |
| Retired | 2 | 0.7 |
| Full time carer | 2 | 0.7 |
| Other | 12 | 4.1 |
| Highest Education Level | | |
| School Certificate | 29 | 9.9 |
| Trade Certificate | 17 | 5.8 |
| Higher School Certificate | 140 | 47.8 |
| Bachelor Degree | 59 | 20.1 |
| Postgraduate Degree | 41 | 14.0 |
| Doctorate | 7 | 2.4 |
| Country of Origin | | |
| Australia | 226 | 77.1 |
| New Zealand | 2 | 0.7 |
| Asia | 11 | 3.8 |
| Europe | 4 | 1.4 |
| United Kingdom | 7 | 2.4 |
| North American | 2 | 0.7 |
| South American | 2 | 0.7 |
| Middle East | 11 | 3.8 |
| Africa | 3 | 1.0 |
| Other | 25 | 8.5 |

## 2.2. Measures

**2.2.1. Generalised Anxiety Disorder Dimensional Scale.** The Generalised Anxiety Disorder Dimensional Scale (GAD-D) [9]) is a 10-item measure that assesses the frequency of symptoms relating to fear, anxiety, and worry, as well as the frequency of behaviours intended to avoid worry and escape feared situations. All items are rated on a 5-point Likert scale ranging from zero (never) to four (all of the time), resulting in a total score ranging between 0 and 40. Previous studies have established excellent internal reliability for the scale ($\alpha$ = .92; [9]).

**2.2.2. Generalised Anxiety Disorder Scale– 7 item.** The Generalised Anxiety Disorder Scale– 7 item (GAD-7) [21], is a scale measuring GAD symptomology. Participants rate the frequency with which they experienced seven symptoms of GAD during the past two weeks, on a 4-point Likert scale ranging from 0 (*not at all*) to 3 (*nearly every day*). Total scores range between 0 and 21. A score of 10 or more on the GAD-7 was established as the cut-point with an optimal balance of sensitivity (89%) and specificity (82%). Good internal reliability has been established in previous studies ($\alpha$ = .88) [22]. Cronbach's $\alpha$ in the present study was .91.

**2.2.3. Panic Disorder Severity Scale–Self Report Version.** The Panic Disorder Severity Scale–Self Report Version (PDSS-SR) [23] is a seven item measure that assesses the frequency of panic attacks, anxiety, and avoidance. Participants rate the severity of panic disorder symptoms on a scale from 0 (*none*) to 4 (*extreme*), resulting in a total score ranging from 0 to 28. Houck et al. [23] demonstrated excellent internal reliability for the scale, Cronbach's $\alpha$ = .91. Cronbach's $\alpha$ in the present study was .92.

## 2.3. Procedure

Data was collected within a larger research study. Participants were recruited via social media, advertisements on community noticeboards and via email. Recruitment source was not monitored in this study. Part 1 consisted of an online questionnaire, that took approximately 20 minutes to complete. Participants viewed an information sheet and consent form, before moving on to a demographic questionnaire, the GAD-D, the GAD-7, and the PDSS-SR. Upon completion of Part 1, participants were invited to participate in Part 2 of this study to assess test-retest reliability of the GAD-D. Those who agreed were asked to input an anonymous, unique identification code to link their response from Part 1 to Part 2. Those participants were emailed by the researchers two weeks after completing Part 1, inviting them to complete the measures again. Part 2 of this study was requested to be undertaken with a two-week time interval. All data was collected via Qualtrics. Data is available from the lead investigator upon reasonable request.

## 2.4. Statistical analysis

Statistical analyses were conducted using IBM SPSS version 26 for Mac and R version 4.02 for Mac. To determine whether the GAD-D demonstrates a unidimensional factor structure, confirmatory factor analysis (CFA) was conducted using the R package "lavaan" version 0.6–7 [24]. The use of this analysis is consistent with that used in the existing literature [16, 17, 30, 31]. The weighted least squares means and variance adjusted (WLSMV) estimation method was used; this method does not assume that variables are normally distributed, and is considered the best option for ordinal data [25]. The following fit-indices were calculated: comparative fit index (CFI), Tucker-Lewis index (TLI), and root mean square error of approximation (RMSEA). A CFI and TLI value of 0.90 or higher is considered an acceptable fit, and 0.95 or higher is considered a good fit [26]. A RMSEA value of 0.08 or less is considered an acceptable fit, and lower than 0.05 is considered a good fit [26]. To evaluate internal consistency, Cronbach's $\alpha$ was calculated for the GAD-D. Convergent validity was examined by calculating

Spearman's rank order correlation ($r_s$) between the total score on the GAD-D and the GAD-7. Consistent with previous studies of the DSM-5 dimensional anxiety scales [9, 10], discriminant validity was defined as being demonstrated by a significantly greater correlation between the GAD-D and a previously validated measure of the same construct than between the GAD-D and a previously validated measure of a distinct construct. Discriminant validity was examined by comparing $r_s$ values of the GAD-D and GAD-7 with $r_s$ values between the GAD-D and the PDSS-SR, a conceptually distinct measure. Correlations were compared using Steiger's [27] modification of Dunn and Clark's [28] $z$ using average correlations, available within the R package cocor version 1.1–3 [29]. Strength of correlation coefficient effect sizes were interpreted according to Cohen [30], where .10 is "small," .30 is "medium," and .50 and above is "large." Test-retest reliability was evaluated by calculating the intra-class correlational coefficient (ICC) between total score on the GAD-D at Part 1 and Part 2. A two-way mixed effects model with absolute agreement was used to calculate the ICC [31]. Consistent with previous studies of the DSM-5 dimensional anxiety scales, an ICC above .70 was considered adequate [9]. Power analysis indicates that with a medium effect size, alpha of .05 and power of .80, 181 participants were required to assess factor analysis [32]. This figure is also consistent with Consensus-based standards for the selection of health measurement instruments guidelines [33], thus the current study was adequately powered for the factor analysis. According to Bujan and Baharum [34], with two observations per participant, an ICC greater than 0.5, an α of 0.05, and power of 0.80, 22 participants are required to estimate the value of ICC for test-retest reliability.

## 3. Results

### 3.1. Missing data

Missing data across the variables ranged from 3.3% to 21.2%. Little's MCAR test, designed to test for patterns in missing data, indicated that the data was missing completely at random, $\chi 2(720) = 678.01$, $p = .87$. Listwise deletion of cases with missing data is appropriate where data is missing completely at random and adequate power is retained [35]. As the data were missing completely at random, 61 participants with missing data and one participant that was identified by the researchers as having completed the questionnaires twice were excluded from further analysis.

### 3.2. Assumption testing

Assumptions of normality, linearity, and homoscedasticity were assessed. Visual inspection of histograms indicated that each of the variables of interest deviated significantly from normal distribution, with each demonstrating positive skew. No outliers (operationalised as > 3.29 standard deviations above the mean) were detected for scores on the GAD-D or GAD-7. However, three outliers were identified on the PDSS-SR. To reduce bias introduced by these outliers and the positive skew on each variable, Spearman's rank-order coefficient was used when calculating correlations, and the WLSMV estimation method was used to conduct CFA.

### 3.3. Descriptive statistics

Descriptive statistics for the sample are reported in Table 2. Of the 293 participants included in Part 1, 137 (46.76%) participants scored 0–10 on the GAD-D, 102 (34.81%) scored 11–20, 40 (13.65%) scored 21–30, and 14 (4.78%) scored 31–40.

### 3.4. Attrition

At the conclusion of Part 1, 140 (48%) individuals expressed interest in completing Part 2 of the study. Of these, 21 participants provided complete data for Part 2, after one email follow-

**Table 2. Descriptive statistics for the measures (N = 293).**

| Measure | M (SD) | Observed range | Possible range |
|---|---|---|---|
| GAD-D | 12.60 (9.26) | 0–40 | 0–40 |
| GAD-7 | 8.06 (5.43) | 0–21 | 0–21 |
| PDSS-SR | 3.71 (4.61) | 0–24 | 0–28 |

*Note.* GAD-D = Generalised Anxiety Disorder Dimensional Scale; GAD-7 = Generalised Anxiety Disorder Scale– 7 Item; PDSS-SR = Panic Disorder Severity Scale–Self Report.

up. These participants did not differ from those who completed only Part 1 in terms of gender ($\chi^2$[2, N = 293] = 0.67, $p > .05$) or level of education ($\chi^2$[5, N = 293] = 1.88, $p > .05$). The two groups did not differ significantly on GAD-D scores, $t(25.488) = 0.87$, $p > .05$.

### 3.5. Factor structure

The results of the CFA are shown in Table 3. The CFI, TLI, and RMSEA fell below the acceptable threshold, indicating a poor fit for the single factor model. Post-hoc inspection of modification indices revealed a strong local dependency between items 6 and 7. Another CFA was conducted, including the correlation between errors of items 6 and 7 in the model. The CFI, TLI, and RMSEA were all acceptable (see Table 3). The correlation coefficient of the local dependency estimates between the errors of items 6 and 7 was 0.31 ($p < .001$). The unstandardized and standardised regression weights for each item in the best-fit model are shown in Table 4. Correlation coefficients between each of the GAD-D items are shown in Table 5.

### 3.6. Reliability

Cronbach's α for the GAD-D was .94 at Part 1 and .91 at Part 2. These results indicate excellent internal consistency. The ICC between GAD-D scores at Part 1 and Part 2 was .85 ($p < .001$; *95% CI = .66-.94*). This strong, positive correlation indicates good test-retest reliability.

### 3.7. Validity

A strong, positive correlation was found between the GAD-D and GAD-7, $r_s = .77$ ($p < .001$), indicating convergent validity. A strong, positive correlation was also found between the GAD-D and PDSS-SR, $r_s = .63$ ($p < .001$). These correlations were compared using Steiger's (1980) method for comparing elements of a correlation matrix. The correlation between the GAD-D and GAD-7 was significantly greater than the correlation between the GAD-D and PDSS-SR, $z(290) = 4.79$ ($p < .001$, two-tailed), providing evidence for discriminant validity.

**Table 3. Results of confirmatory factor analysis.**

| Instrument | χ2 (df) | CFI | TLI | RMSEA (90% CI) |
|---|---|---|---|---|
| *CFA* | | | | |
| GAD-D | 116.441 (35), $p < .001$ | 0.895 | 0.865 | 0.089 (0.072–0.107) |
| *CFA (local dependency items 6 and 7)* | | | | |
| GAD-D | 90.133(34), $p < .001$ | 0.927 | 0.904 | 0.075 (0.057–0.094) |

*Note.* CFA = Confirmatory Factor Analysis; CFI = Comparative Fit Index, TLI = Tucker Lewis Index; RMSEA = Root Mean Square Error of Approximation; GAD-D = Generalised Anxiety Disorder Dimensional Scale.

**Table 4. Confirmatory factor analysis factor loadings for GAD-D items (N = 293).**

| Item | Unstandardised | Standardised | *p*-value | Squared multiple correlations |
|------|---------------|--------------|-----------|-------------------------------|
| Item 1 | 1.00 | 0.81 | < .001 | 0.66 |
| Item 2 | 0.92 | 0.81 | < .001 | 0.65 |
| Item 3 | 0.97 | 0.78 | < .001 | 0.60 |
| Item 4 | 0.90 | 0.81 | < .001 | 0.65 |
| Item 5 | 0.97 | 0.82 | < .001 | 0.67 |
| Item 6 | 0.97 | 0.79 | < .001 | 0.62 |
| Item 7 | 0.93 | 0.75 | < .001 | 0.57 |
| Item 8 | 1.08 | 0.83 | < .001 | 0.69 |
| Item 9 | 0.88 | 0.65 | < .001 | 0.43 |
| Item 10 | 0.73 | 0.62 | < .001 | 0.38 |

*Note*. GAD-D = Generalised Anxiety Disorder Dimensional Scale

## 4. Discussion

The GAD-D is a dimensional measure of GAD symptom severity recently developed by the DSM-5 Anxiety, Obsessive-Compulsive Spectrum, Posttraumatic, and Dissociative Disorder Work Group. The scale is quick to administer and purports to provide a dimensional assessment of GAD as an alternative to the prevailing categorical approach. The aim of the present study was to evaluate the psychometric properties of the scale in an Australian community sample. The GAD-D was found to have a unidimensional factor structure, excellent internal consistency, and acceptable test-retest reliability. Convergent and discriminant validity were demonstrated.

Results of the study suggest that the GAD-D has a unidimensional factor structure, consistent with previous examinations of the scale's dimensionality [10, 14]. This indicates that the items of the GAD-D measure one underlying construct, i.e., the severity of GAD symptomology. The present study is the first to use the English language version of the GAD-D to examine its factor structure. However, the fit indices indicated only an adequate model fit. A larger sample may have resulted in better model fit, as was observed in previous studies of the GAD-D that utilised CFA [10, 14].

Consistent with the findings of DeSousa et al. [14] in their Brazilian sample, model fit was notably improved when the local dependency between items 6 and 7 was included in the

**Table 5. Spearman rank-order correlations between GAD-D items (N = 293).**

| Item | 1 | 2 | 3 | 4 | 5 | 6 | 7 | 8 | 9 | 10 |
|------|---|---|---|---|---|---|---|---|---|----|
| 1 | - | | | | | | | | | |
| 2 | .66 | - | | | | | | | | |
| 3 | .67 | .63 | - | | | | | | | |
| 4 | .64 | .62 | .60 | - | | | | | | |
| 5 | .61 | .69 | .58 | .70 | - | | | | | |
| 6 | .61 | .57 | .57 | .63 | .59 | - | | | | |
| 7 | .59 | .51 | .54 | .62 | .56 | .79 | - | | | |
| 8 | .61 | .66 | .58 | .62 | .66 | .71 | .69 | - | | |
| 9 | .45 | .55 | .48 | .43 | .53 | .50 | .45 | .60 | - | |
| 10 | .45 | .39 | .41 | .50 | .46 | .43 | .50 | .46 | .51 | - |

*Note*. All correlations significant at *p* < .001.

model. Item 6 refers to avoiding situations about which the individual worries, while item 7 refers to leaving or participating minimally in situations about which the individual worries. In the present study, the correlation between these two items was the largest correlation found between any two GAD-D items. Of the most prevalent DSM-5 anxiety disorders (including social phobia, specific phobia, panic disorder, and agoraphobia), GAD is unique in that avoidance behaviour is not included as a criterion for diagnosis. However, the loading of these two items on the general factor were comparable to the other items on the GAD-D. This may suggest that avoidance and escape behaviours are typically observed in individuals that have GAD, despite their exclusion from the DSM-5 criteria. This would align particularly with models of GAD that highlight the maintaining role of avoidance; for example, those proposed by Borkovec [16] and Dugas et al. [36]. Future research may wish to further explore this.

Notably, although all items demonstrated adequate factor loadings in the present study, items 9 and 10 had a lower loading onto the general factor than the other items of the scale. These items refer to [9] seeking reassurance from others due to worries, and [10] needing help to cope with worries from alcohol, medication, superstitious objects, or other people. This result was inconsistent with the factor loadings reported by DeSousa et al.[14] in their Brazilian sample. One reason for this may be a cultural difference between Australians and Brazilians: Australians are perhaps typically more individualistic than Brazilians [37] and therefore less likely to seek reassurance from others or enlist their help to cope with worries. However, there is little other data to support this theory, as only two studies conducted CFA on the GAD-D, one of which did not report factor loadings [10]. Therefore, it is difficult to draw conclusions about why this pattern emerged. Researchers validating the GAD-D in the future should conduct CFA where sample size allows, providing factor loadings for each item.

Consistent with previous studies [9, 11, 12], good internal consistency was found for the GAD-D in both the original sample and in the test-retest condition. The test-retest reliability of the scale was also acceptable, indicated by a strong, positive ICC between GAD-D scores at Part 1 and Part 2. This replicates in an Australian sample the results reported by Knappe et al. [11] and LeBeau et al. [9]. Like these two studies, participants completing the GAD-D were asked to report GAD symptoms over the past month. DeSousa et al. [14] asked participants to report symptoms over the past week and found an ICC below the .70 threshold for acceptable test-retest reliability. Notably, the current publicly available version of the GAD-D also asks the individual to report symptoms over the past week. The present study contributes further evidence that the "past month" version of the scale is more reliable across time. Clinicians using the currently available version of the GAD-D to monitor symptoms on a regular basis should consider these results, noting that fluctuations of symptoms in any one week may not be indicative of overall GAD symptomology; GAD currently requires a history of six months of symptoms experienced more days than not for diagnosis [1].

Convergent validity was established by a strong, positive correlation between the GAD-D and the GAD-7, a widely used screening tool for GAD with an established cut-point for diagnosis [21]. This finding was consistent with all previous evaluations of the GAD-D. Due to this high correlation, future studies may consider the incremental validity of the measure, given that the GAD-D is three items longer than the GAD-7. The advantage of the GAD-D may instead lie in its format, which is consistent with the other DSM-5 dimensional anxiety scales, allowing a diagnostic profile to be constructed across anxiety disorders. Importantly, the GAD-D includes measures of avoidance and escape behaviours, which are typically observed in GAD but not included in the GAD-7 or the DSM-5 criteria. As avoidance is widely considered a perpetuating factor in GAD [16, 36], the GAD-D may provide incrementally more salient information to clinicians than the GAD-7.

Discriminant validity was established by comparing the correlation between the GAD-D and GAD-7 with the correlation between the GAD-D and a measure of panic disorder symptomology, the PDSS-SR. While both correlations were large and positive, the GAD-D and GAD-7 demonstrated a significantly greater correlation, consistent with LeBeau et al. [9]. This result indicates a large degree of overlap and comorbidity between GAD and panic disorder, a trend observed across anxiety disorders [38]. Therefore, future studies should test the specificity and sensitivity of the GAD-D to ensure that diagnosis is not too broadly applied.

One advantage of the present study is that the full range of response severity was observed with at least one participant having the minimum and maximum possible score on the GAD-D and GAD-7. This indicates that the sample likely included individuals with GAD and other anxiety disorders, as well as subclinical and healthy individuals from the community. The sample was also broad in terms of marital status, employment status, and highest education level. Further, 27.3% of participants were born overseas, approximately consistent with the Australian Bureau of Statistics [39] report stating that 29.7% of the Australian population were born overseas.

Several limitations of the present study should be considered. First, only a small percentage of participants (7.12%) completed Part 2 of the study. This is likely due to only one follow up email being sent to participants who expressed interest in completing this part of the study. Whilst close to recommended sample size for test re-test reliability, this small sample size may limit the generalisability of the test-retest reliability found in the present study and findings should be considered preliminary. Importantly though, other studies that have examined test-retest reliability of the GAD-D have found similar results [9, 11]. Further, the time between completion of Part 1 and Part 2 for test-retest was not strictly adhered to by participants. Future studies aiming to achieve greater generalisability should restrict or control for time elapsed between Part 1 and Part 2 for each participant. It is possible that individuals who completed Part 2 were more invested in the content of the study and may represent a distinct population compared to the overall sample. To prevent this, it may also help to provide a monetary incentive for individuals to complete both parts. Future research may also wish to use more follow up emails to increase participation in Part 2.

Although a full range of responses were recorded for each GAD measure, mean scores on the GAD-D were elevated in comparison to those reported in other studies. A similar pattern was observed regarding scores on the GAD-7 and PDSS-SR where these were utilised [9, 14]. For example, while the present study had a mean of 12.60 ($SD$ = 9.26) on the GAD-D, the sample of German treatment-seeking individuals had a mean of 11.8 ($SD$ = 9.9), indicating that the present community sample had a similar or even greater level of GAD or other anxiety symptomology than a sample of treatment-seeking individuals. Other studies of community samples reported lower mean scores on the GAD-D [9, 14]. While this may be in line with the elevated rates of GAD in the Australian community [2], it is possible that the study attracted a population affected by anxiety and thus especially concerned with research in this area. This effect may have been further exacerbated by the predominantly female sample, as females are more likely on average to experience GAD [40]. These factors may have led to a sample that was not representative of the Australian community.

A further limitation regards the validity of responses, given the extreme scores endorsed by a small percentage of participants on the measures of interest. In the present study, a small number of individuals endorsed items resulting in a maximum score on one or more measures. While visual inspection of overall scores did not reveal a pattern of repetitive responding across all items for any of the study participants, the incorporation of attention check items would allow for a more comprehensive assessment of this possibility. Fortunately, in the present study, the presence of extreme scores is unlikely to have affected any of the key findings, as

non-parametric methods were used to calculate correlation coefficients, and the CFA estimation method chosen is known to be robust against outliers and violation of normality [26].

Overall, the present study replicates the small but growing body of literature suggesting that the GAD-D is a reliable and valid measure and is therefore appropriate for use in establishing baseline severity and monitoring GAD symptomology in the Australian population. Future research should analyse the sensitivity and specificity of the GAD-D in an Australian sample to establish a cut-off point and allow Australian clinicians to also use the GAD-D as a screening tool, using a clinical sample of individuals who have received diagnosis via structured interview. Finally, future research should aim to establish the psychometric properties of the other DSM-5 dimensional anxiety scales. If shown to be valid and reliable, the scales given as a battery would offer an efficient and consistent way for clinicians to profile individuals across anxiety disorders, potentially leading to more appropriately targeted treatment.

## Supporting information

**S1 File. Key statistical output.**
(DOCX)

## Author Contributions

**Conceptualization:** Bethany Wootton, Karen Moses.

**Data curation:** David Groves, Theodora Binasis, Karen Moses.

**Formal analysis:** David Groves.

**Investigation:** Karen Moses.

**Methodology:** Bethany Wootton, Karen Moses.

**Project administration:** David Groves, Theodora Binasis, Karen Moses.

**Software:** Karen Moses.

**Supervision:** Bethany Wootton, Karen Moses.

**Writing – original draft:** David Groves.

**Writing – review & editing:** David Groves, Theodora Binasis, Bethany Wootton, Karen Moses.

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
