## [Decision Letter · Decision Letter 0]

1 Jul 2022

PONE-D-21-36347

Psychometric properties of the Generalised Anxiety Disorder Dimensional Scale in an Australian sample

PLOS ONE

Dear Dr. Moses,

Thank you for submitting your manuscript to PLOS ONE. After careful consideration, we feel that it has merit but does not fully meet PLOS ONE’s publication criteria as it currently stands. Therefore, we invite you to submit a revised version of the manuscript that addresses the points raised during the review process.

In particular, please address the methodological comments raised by reviewer 1.

We look forward to receiving your revised manuscript.

Kind regards,

Yann Benetreau, PhD

Division Editor (Staff Editor)

PLOS ONE

Journal Requirements:

3. We note that your Data Availability statement states the following: "Data cannot be shared publicly because it remains under use by students and staff of the Western Sydney University."

PLOS journals require that all data presented in the study be made publicly available at or before the time of publication. If there are legal or ethical restrictions on the data being made publicly available, such as IRB restriction or patient confidentiality, authors must provide a way for fellow researchers to access the data.

Before we can proceed, please clarify the following:

a. Are there legal or ethical restrictions being placed upon the data? If so, please explain them in detail (e.g., data contain potentially identifying or sensitive patient information, data are owned by a third-party organization, etc.) and who has imposed them (e.g., a Research Ethics Committee or Institutional Review Board, etc.).

b. If there are no legal or ethical restrictions, please upload the data as a Supporting Information file or to a recommended stable public repository. (https://journals.plos.org/plosone/s/recommended-repositories)

c. Please note that PLOS does not allow authors to be the sole contact for data inquiries. If the data is only available upon request, please provide contact information, such as an email address, for a non-author, institutional point of contact (such as an IRB or ethics committee contact) who can field data inquiries from fellow researchers. If the data contact is an individual, please provide their title and relationship to the data as well.

Reviewers' comments:

Reviewer's Responses to Questions

**Comments to the Author**

1. Is the manuscript technically sound, and do the data support the conclusions?

Reviewer #1: No

Reviewer #2: Yes

2. Has the statistical analysis been performed appropriately and rigorously? 

Reviewer #1: No

Reviewer #2: Yes

3. Have the authors made all data underlying the findings in their manuscript fully available?

Reviewer #1: No

Reviewer #2: Yes

4. Is the manuscript presented in an intelligible fashion and written in standard English?

Reviewer #1: No

Reviewer #2: Yes

5. Review Comments to the Author

Reviewer #1: Psychometric properties of the Generalised Anxiety Disorder Dimensional Scale in an Australian sample

1- Introduction: It is very long, tedious and contains unnecessary information.

2- This questionnaire assess Anxiety Disorder but there is no mention of having psychiatric disorders, taking certain medications, etc. in patients. Only demographic characteristic has been presented.

3- Willing to participation is not as inclusion criteria

4- In construct validity, why Exploratory Factor Analysis (EFA) was not done?

5- Discriminate validity indicates how well an instrument differentiates between different groups who differ in some characteristics. In this study in abstract, there is divergent and in manuscript is discriminate validity. "The correlation between the GAD-D and GAD-7 was significantly greater than the correlation between the GAD-D and PDSS-SR, z(290) = 4.79 (p < .001, two-tailed), providing evidence for discriminate validity". What is the meaning of authors of z(290) = 4.79? Is that correlation between GAD-D and PDSS-SR?

6- The manuscript repeatedly mentions Part 1 and 2 and Time 1 and 2. If time 1 and 2 referred to test retest time reliability, what is the meaning of parts 1 and 2?

7- The interval time for assess test retest reliability is not mentioned.

8- The manner of the method section is chaotic.

9- It's mention that" The time between completion of Part 1 and Part 2 was not controlled or monitored". Why? This is very important and critical.

Reviewer #2: ------------------------------------------------------------------------------------------------------------------------------------------------------------------------------------------------------------

6. PLOS authors have the option to publish the peer review history of their article (what does this mean?). If published, this will include your full peer review and any attached files.

Reviewer #1: No

Reviewer #2: No

---

## [Author Response · Author response to Decision Letter 0]

14 Sep 2022

Reviewer 1 

1- Introduction: It is very long, tedious and contains unnecessary information.

The authors thank the reviewer for this feedback. The introduction has now been revised to increase succinctness and clarity. Please see pages 3-5 of the manuscript. 

2- This questionnaire assess Anxiety Disorder but there is no mention of having psychiatric disorders, taking certain medications, etc. in patients. Only demographic characteristic has been presented.

The authors thank the reviewer for this feedback. As a non-clinical community sample has been utilised, psychiatric history was not sought from participants. This has now been clearly referenced. Please see page 6 of the manuscript. 

3- Willing to participation is not as inclusion criteria

The authors thank the reviewer for this feedback. The authors have reviewed the manuscript to ensure that willingness to participate has not been included as an inclusion criteria. 

4- In construct validity, why Exploratory Factor Analysis (EFA) was not done?

The authors thank the reviewer for this query. Given previous psychometric evaluations of the DSM 5 Dimensional Scales internationally, confirmatory factor analysis was deemed most appropriate to undertake in this study. Importantly, this approach is also consistent with existing literature, including Lebeau et al., 2012, Beesdo-Baum et al., 2012, Macfarlane et al., 2020 and Russell et al., 2020. This has now also been referenced in the manuscript. Please see page 7. 

5- Discriminate validity indicates how well an instrument differentiates between different groups who differ in some characteristics. In this study in abstract, there is divergent and in manuscript is discriminate validity. "The correlation between the GAD-D and GAD-7 was significantly greater than the correlation between the GAD-D and PDSS-SR, z(290) = 4.79 (p < .001, two-tailed), providing evidence for discriminate validity". What is the meaning of authors of z(290) = 4.79? Is that correlation between GAD-D and PDSS-SR?

The authors thank the reviewer for this observation. Incorrect reference to divergent validity has now been removed. Please see page 2 of the manuscript. 

6- The manuscript repeatedly mentions Part 1 and 2 and Time 1 and 2. If time 1 and 2 referred to test retest time reliability, what is the meaning of parts 1 and 2?

The authors thank the reviewer for this observation. All references to Time 1 and Time 2 have now been changed to Part 1 and Part 2 respectively. This has been changed throughout the manuscript, highlighted in red. 

7- The interval time for assess test-retest reliability is not mentioned.

This has now been included. Please see page 7 of the manuscript. 

8- The manner of the method section is chaotic.

The authors thank the reviewer for this feedback. The method section has now been revised. Please see pages 5-8 of the manuscript. 

9- It's mention that" The time between completion of Part 1 and Part 2 was not controlled or monitored". Why? This is very important and critical.

The authors thank the reviewer for this feedback. Whilst completion time between Part 1 and Part 2 was monitored, it was not strictly adhered to by participants. This section has been reworded for clarity. Please see page 14 of the manuscript.

Reviewer 2 

N/A

---

## [Decision Letter · Decision Letter 1]

23 Feb 2023

PONE-D-21-36347R1Psychometric properties of the Generalised Anxiety Disorder Dimensional Scale in an Australian samplePLOS ONE

Dear Dr. Moses,

Thank you for submitting your manuscript to PLOS ONE. After careful consideration, we feel that it has merit but does not fully meet PLOS ONE’s publication criteria as it currently stands. Therefore, we invite you to submit a revised version of the manuscript that addresses the points raised during the review process.

We look forward to receiving your revised manuscript.

Kind regards,

Alejandro Vega-Muñoz, Ph.D.

Academic Editor

PLOS ONE

Journal Requirements:

Reviewers' comments:

Reviewer's Responses to Questions

**Comments to the Author**

1. If the authors have adequately addressed your comments raised in a previous round of review and you feel that this manuscript is now acceptable for publication, you may indicate that here to bypass the “Comments to the Author” section, enter your conflict of interest statement in the “Confidential to Editor” section, and submit your "Accept" recommendation.

Reviewer #3: (No Response)

Reviewer #4: (No Response)

2. Is the manuscript technically sound, and do the data support the conclusions?

Reviewer #3: Partly

Reviewer #4: Yes

3. Has the statistical analysis been performed appropriately and rigorously? 

Reviewer #3: Yes

Reviewer #4: Yes

4. Have the authors made all data underlying the findings in their manuscript fully available?

Reviewer #3: No

Reviewer #4: No

5. Is the manuscript presented in an intelligible fashion and written in standard English?

Reviewer #3: Yes

Reviewer #4: No

6. Review Comments to the Author

Reviewer #3: I consider this to be a valuable article that presents the psychometric properties of the Generalised Anxiety Disorder Dimensional Scale in an Australian cummunity sample. However, there are a couple of issues to be improved that could help make the manuscript more comprehensive. These topics are described below.

Abstract

“A sample of 293 Australians (72.7% female) aged between 18 and 73 (M = 28.31 years; SD = 12.11 years) were recruited.” a sample was…?

Introduction

1.“The number of participants required for the study was based on the recommendation of

Tabachnick and Fidell(32), who specified that a minimum of 300 participants is adequate for a

psychometric analysis that includes examination of factor structure.”

I believe the authors could provide more information on the sample size planning procedure. Not just indicate a defined number for the sample. For example, see Kelley, K., & Lai, K. (2018). Confirmatory factor models: Power and accuracy for effects of interest. The Wiley handbook of psychometric testing: A multidisciplinary reference on survey, scale and test development, 113-139.

2. “This resulted in a final sample of 293(< 300) participants.” The final sample is smaller than initially proposed.

Methods

1. p.8, about the Generalised anxiety disorder dimensional scale “Cronbach’s α in the present study was .94.” I believe this should be reported in the results

Results

It is not explicitly stated in the manuscript whether the analyzed data are available in any public repository. If not, it would be ideal if the authors would do so.

Discussion

1.p.13 “To the author’s knowledge,…” to the authors' knowledge?

2. “Therefore, future studies evaluating the psychometric properties of the GAD-D should attempt to exceed the minimum 300 participants specified by Tabachnick and Fidell(32)” Again, I believe that the authors could make explicit a sampling procedure based on CFA, especially if the authors suggest that a higher n could guarantee a better model fit.

3. "This may suggest that avoidance and escape behaviours are typically observed in individuals that have GAD, despite their exclusion from the DSM-5 criteria. This would align particularly with models of GAD that highlight the maintaining role of avoidance; for example, those proposed by Borkovec (27) and Dugas et al.(42)" Interesting, future proposed research could add something to this point, considering the assessment (self-report) of avoidance and escape behaviors in clinical population having or not the GAD diagnosis.

4. “Notably, although all items demonstrated adequate factor loadings in the present study, items 9 and 10 had a notably lower loading onto the general factor than the other items of the scale.” Any of the expressions "notably" may be changed.

5. "...indicating that the present community sample had a similar or even greater level of GAD pathology than a sample of treatment-seeking individuals." It would not assert that it is GAD pathology if there is no clinical diagnosis to support it, but rather related symptoms or or anxiety problems.

6. "Future research should analyse the sensitivity and specificity of the GAD-D in an Australian sample to establish a cut-off point and allow Australian clinicians to also use the GAD-D as a screening tool". I believe that it should be emphasized that in order to achieve "cut-off points" it is necessary to include in the sample a properly diagnosed clinical population.

Reviewer #4: I congratulate authors for revising their manuscript. Manuscript is well written. I have a few suggestions.

1. Authors should provide what methods they used for data collection. While authors have mentioned an online questionnaire was used, there is no mention of website or software used for online survey.

2. While 140 (48%) individuals expressed interest in completing Part 2, only 21 participants provided complete data for Part 2. Any specific reason for very small retention? Does authors sent any reminder to those who showed interest for participation in part 2?

3. Sample size for retest analysis is too small. I would suggest to follow-up with those who showed interest to participate in part 2. Alternatively, authors may conduct additionally survey, which include at least 5o subjects in retest survey.

7. PLOS authors have the option to publish the peer review history of their article (what does this mean?). If published, this will include your full peer review and any attached files.

Reviewer #3: No

Reviewer #4: **Yes: **Shahnawaz Anwer

---

## [Author Response · Author response to Decision Letter 1]

4 Apr 2023

Response to Reviewers 

Journal Requirements:

- The reference list has now been reviewed for accuracy and completeness. One omitted reference from the reference list has now been added. Please see page 22 of the revised manuscript.

To the authors knowledge, no retracted articles have been included in this manuscript. Three new references have been added, relevant to reviewer requests. Please see page 22 of the revised manuscript. 

Reviewer #3: 

Abstract

“A sample of 293 Australians (72.7% female) aged between 18 and 73 (M = 28.31 years; SD = 12.11 years) were recruited.” a sample was…?

- The authors thank the reviewer for identifying this error. This has now been rephrased as follows: ‘A sample of 293 Australians (72.7% female) aged between 18 and 73 (M = 28.31 years; SD = 12.11 years) was recruited.’ Please see page 2 of the revised manuscript. 

Introduction

1.“The number of participants required for the study was based on the recommendation of

Tabachnick and Fidell(32), who specified that a minimum of 300 participants is adequate for a

psychometric analysis that includes examination of factor structure.”

I believe the authors could provide more information on the sample size planning procedure. Not just indicate a defined number for the sample. For example, see Kelley, K., & Lai, K. (2018). Confirmatory factor models: Power and accuracy for effects of interest. The Wiley handbook of psychometric testing: A multidisciplinary reference on survey, scale and test development, 113-139.

- The authors thank the reviewer for this feedback. A sampling size procedure has now been listed in the manuscript, which suggests adequate sample size used. This is also consistent with COSMIN guidelines (Terwee et al., 2011). Please see page 9 of the revised manuscript. 

2. “This resulted in a final sample of 293(< 300) participants.” The final sample is smaller than initially proposed.

- Given use of sampling procedure (as recommended above), sample is now considered within appropriate range. 

Methods

1. p.8, about the Generalised anxiety disorder dimensional scale “Cronbach’s α in the present study was .94.” I believe this should be reported in the results

- The authors thank the reviewer for this feedback. This is now recorded in the results section only. Please see page 10 of the revised manuscript. 

Results

It is not explicitly stated in the manuscript whether the analyzed data are available in any public repository. If not, it would be ideal if the authors would do so.

- The authors thank the reviewer for highlighting this omission. As the data used for this project was collected as part of a larger body of research still currently in use, data is not currently available in a public repository, though may be requested from the lead investigator. This has now been added to the manuscripts as follows: ‘Data is available from the lead investigator upon reasonable request. ‘ Please see page 7 of the revised manuscript.

Discussion

1.p.13 “To the author’s knowledge,…” to the authors' knowledge?

- For clarity of language, this has now been removed. Please see page 13 of the revised manuscript. 

2. “Therefore, future studies evaluating the psychometric properties of the GAD-D should attempt to exceed the minimum 300 participants specified by Tabachnick and Fidell(32)” Again, I believe that the authors could make explicit a sampling procedure based on CFA, especially if the authors suggest that a higher n could guarantee a better model fit.

- The authors thank the reviewer again for this recommendation. A revised sampling procedure has now been included, which suggests sample utilised is within appropriate range. Please see page 9 of the revised manuscript.

3. "This may suggest that avoidance and escape behaviours are typically observed in individuals that have GAD, despite their exclusion from the DSM-5 criteria. This would align particularly with models of GAD that highlight the maintaining role of avoidance; for example, those proposed by Borkovec (27) and Dugas et al.(42)" Interesting, future proposed research could add something to this point, considering the assessment (self-report) of avoidance and escape behaviors in clinical population having or not the GAD diagnosis.

- Thank you for this observation. This has now been highlighted as an area of further research in the manuscript. Please see page 12 of the revised manuscript.

4. “Notably, although all items demonstrated adequate factor loadings in the present study, items 9 and 10 had a notably lower loading onto the general factor than the other items of the scale.” Any of the expressions "notably" may be changed.

- Thank you for this observation. Wording has now been changed to the following: ‘Notably, although all items demonstrated adequate factor loadings in the present study, items 9 and 10 had a lower loading onto the general factor than the other items of the scale.’ Please see page 12 of the revised manuscript.

5. "...indicating that the present community sample had a similar or even greater level of GAD pathology than a sample of treatment-seeking individuals." It would not assert that it is GAD pathology if there is no clinical diagnosis to support it, but rather related symptoms or or anxiety problems.

- The authors thank the reviewer for highlighting this important distinction. This has now been corrected in the manuscript as follows: ‘indicating that the present community sample had a similar or even greater level of GAD or other anxiety symptomology than a sample of treatment-seeking individuals.’ Please see page 15 of the revised manuscript.

6. "Future research should analyse the sensitivity and specificity of the GAD-D in an Australian sample to establish a cut-off point and allow Australian clinicians to also use the GAD-D as a screening tool". I believe that it should be emphasized that in order to achieve "cut-off points" it is necessary to include in the sample a properly diagnosed clinical population.

- The authors thank the reviewer for identifying this important point. This has now been reflected in the manuscript as follows: ‘Future research should analyse the sensitivity and specificity of the GAD-D in an Australian sample to establish a cut-off point and allow Australian clinicians to also use the GAD-D as a screening tool, using a clinical sample of individuals who have received diagnosis via structured interview.’ Please see page 16 of the revised manuscript.

Reviewer #4: 

1. Authors should provide what methods they used for data collection. While authors have mentioned an online questionnaire was used, there is no mention of website or software used for online survey.

- The authors thank the reviewer for identifying this omission. This has now been included, as follows: ‘All data was collected via Qualtrics.’ Please see page 7 of the revised manuscript.

2. While 140 (48%) individuals expressed interest in completing Part 2, only 21 participants provided complete data for Part 2. Any specific reason for very small retention? Does authors sent any reminder to those who showed interest for participation in part 2?

- The authors thank the reviewer for this feedback. It has now been highlighted in the manuscript (results section) that only one follow up email was sent to participants. Please see pages 9 and 10 of the revised manuscript.

The following has now also been added to the discussion section: ‘This is likely due to only one follow up email being sent to participants who expressed interest in completing this part of the study ‘and ‘Future research may also wish to use more follow up emails to increase participation in Part 2.’ Please see pages 14 and 15 of the revised manuscript.

3. Sample size for retest analysis is too small. I would suggest to follow-up with those who showed interest to participate in part 2. Alternatively, authors may conduct additionally survey, which include at least 5o subjects in retest survey.

- The authors thank the reviewer for this feedback. We accept that the sample size for re-test analysis is small. Unfortunately, collecting further data is not a possibility. It is noted that some references indicate that our sample size is broadly within range. For example, with two observations per subject, an ICC greater than 0.5, an α of 0.05, and power of 0.80, 22 participants are required to estimate the value of ICC (Bujang, 2017). This has now been noted in the manuscript. However, given this, we emphasise that assessment of re-test reliability is preliminary in nature within the manuscript. Please see pages 9 and 15 of the revised manuscript.

---

## [Decision Letter · Decision Letter 2]

27 Apr 2023

PONE-D-21-36347R2Psychometric properties of the Generalised Anxiety Disorder Dimensional Scale in an Australian samplePLOS ONE

Dear Dr. Moses,

Thank you for submitting your manuscript to PLOS ONE. After careful consideration, we feel that it has merit but does not fully meet PLOS ONE’s publication criteria as it currently stands. Therefore, we invite you to submit a revised version of the manuscript that addresses the points raised during the review process.

We look forward to receiving your revised manuscript.

Kind regards,

Alejandro Vega-Muñoz, Ph.D.

Academic Editor

PLOS ONE

Journal Requirements:

Reviewers' comments:

Reviewer's Responses to Questions

**Comments to the Author**

1. If the authors have adequately addressed your comments raised in a previous round of review and you feel that this manuscript is now acceptable for publication, you may indicate that here to bypass the “Comments to the Author” section, enter your conflict of interest statement in the “Confidential to Editor” section, and submit your "Accept" recommendation.

Reviewer #3: All comments have been addressed

Reviewer #4: (No Response)

2. Is the manuscript technically sound, and do the data support the conclusions?

Reviewer #3: Yes

Reviewer #4: Yes

3. Has the statistical analysis been performed appropriately and rigorously? 

Reviewer #3: Yes

Reviewer #4: Yes

4. Have the authors made all data underlying the findings in their manuscript fully available?

Reviewer #3: Yes

Reviewer #4: No

5. Is the manuscript presented in an intelligible fashion and written in standard English?

Reviewer #3: Yes

Reviewer #4: Yes

6. Review Comments to the Author

Reviewer #3: (No Response)

Reviewer #4: I congratulate authors for revising their manuscript incorporating the reviewer comments. However, I still have some minor comments as follow:

1. In the last paragraph of introduction section, author stated that "The following will be investigated: 1) factor structure; 2) reliability; and 3) convergent and discriminant". I think author should use a past tense rather using future tense.

2. It is hypothesized that the psychometric properties of this scale will be consistent with ...?

3. Please explain GAD-D before using abbreviation.

4. This use of this analysis is consistent with that used in the existing literature (16, 17, 30,..)? Please check this sentence

5. Please explain Little’s MCAR test?

6. More than 50% citations are 10 years older. Please replace some of them with more recent citations.

7. PLOS authors have the option to publish the peer review history of their article (what does this mean?). If published, this will include your full peer review and any attached files.

Reviewer #3: No

Reviewer #4: No

---

## [Author Response · Author response to Decision Letter 2]

8 May 2023

Response to Reviewers 

Reviewer #3: (No Response)

Reviewer #4: I congratulate authors for revising their manuscript incorporating the reviewer comments. However, I still have some minor comments as follow:

1. In the last paragraph of introduction section, author stated that "The following will be investigated: 1) factor structure; 2) reliability; and 3) convergent and discriminant". I think author should use a past tense rather using future tense.

The authors thank the reviewer for this observation. This has now been corrected as follows: ‘The following were investigated: 1) factor structure; 2) reliability; and 3) convergent and discriminant validity.’ Please see page 5 of the manuscript. 

2. It is hypothesized that the psychometric properties of this scale will be consistent with ...?

The authors thank the reviewer for this feedback. This has now been corrected to as follows: ‘It was hypothesised that the psychometric properties of this scale will be consistent with previous evaluations (11, 13, 14, 16, 23) . Specifically, the GAD-D is expected to demonstrate a unidimensional factor structure, excellent internal consistency, good test-retest reliability, convergent and discriminant validity.’ Please see page 5 of the manuscript. 

3. Please explain GAD-D before using abbreviation.

The authors thank the reviewer for this observation. This has now been corrected. Please see page 4 of the manuscript. 

4. This use of this analysis is consistent with that used in the existing literature (16, 17, 30,..)? Please check this sentence

The authors thank the reviewer for this observation. This has now been corrected as follows: ‘The use of this analysis is consistent with that used in the existing literature (16, 17, 30, 31).’ Please see page 7 of the manuscript. 

5. Please explain Little’s MCAR test?

The authors thank the reviewer for highlighting this omission. This has now been added as follows: ‘Little’s MCAR test, designed to test for patterns in missing data, indicated that the data was missing completely at random, χ2(720) = 678.01, p = .87.’ Please see page 9 of the manuscript. 

6. More than 50% citations are 10 years older. Please replace some of them with more recent citations.

The authors thank the reviewer for highlighting this. Prior to this feedback, 42 references were included. Of these, 5 have been replaced with references within the last ten years, three have been removed and one has been added. This results in 40 references at this submission. Of these, 15 are 10 years or older, resulting in only 37.5% of references being 10 years or older. Please see page 17 of the manuscript.

---

## [Decision Letter · Decision Letter 3]

22 May 2023

Psychometric properties of the Generalised Anxiety Disorder Dimensional Scale in an Australian sample

PONE-D-21-36347R3

Dear Dr. Moses,

We’re pleased to inform you that your manuscript has been judged scientifically suitable for publication and will be formally accepted for publication once it meets all outstanding technical requirements.

Kind regards,

Alejandro Vega-Muñoz, Ph.D.

Academic Editor

PLOS ONE

Additional Editor Comments (optional):

Reviewers' comments:

Reviewer's Responses to Questions

**Comments to the Author**

1. If the authors have adequately addressed your comments raised in a previous round of review and you feel that this manuscript is now acceptable for publication, you may indicate that here to bypass the “Comments to the Author” section, enter your conflict of interest statement in the “Confidential to Editor” section, and submit your "Accept" recommendation.

Reviewer #4: All comments have been addressed

2. Is the manuscript technically sound, and do the data support the conclusions?

Reviewer #4: Yes

3. Has the statistical analysis been performed appropriately and rigorously? 

Reviewer #4: Yes

4. Have the authors made all data underlying the findings in their manuscript fully available?

Reviewer #4: Yes

5. Is the manuscript presented in an intelligible fashion and written in standard English?

Reviewer #4: Yes

6. Review Comments to the Author

Reviewer #4: (No Response)

7. PLOS authors have the option to publish the peer review history of their article (what does this mean?). If published, this will include your full peer review and any attached files.

Reviewer #4: No

---

## [Editor Report · Acceptance letter]

29 May 2023

PONE-D-21-36347R3 

Psychometric properties of the Generalised Anxiety Disorder Dimensional Scale in an Australian sample 

Dear Dr. Moses:

I'm pleased to inform you that your manuscript has been deemed suitable for publication in PLOS ONE. Congratulations! Your manuscript is now with our production department. 

Kind regards, 

on behalf of

Dr. Alejandro Vega-Muñoz 

Academic Editor

PLOS ONE